

# Environmental fate of the anti-parasitic ivermectin in an aquatic micro-ecological system after a single oral administration

Di Wang[1], Bing Han[2], Shaowu Li[1], Yongsheng Cao[1], Xue Du[1] and Tongyan Lu[1]

[1] Department of Fish Diseases, Heilongjiang River Fisheries Research Institute, Chinese Academy of Fishery Sciences, Harbin/Heilongjiang, China
[2] Department of Pharmacology, School of Medicine, Southeast University, Nanjing/Jiangsu, China

## ABSTRACT

**Background**. Ivermectin (IVM) has been widely used in the aquaculture industry since its efficacy against parasites. However, the degradation of IVM was very slow in aquatic environments and the environmental fate of IVM in a complete aquatic system was still not clear. Therefore, comparable studies in a complete aquatic system were merited and helped to elucidate the environmental fate and effects of IVM.

**Methods**. An aquatic micro-ecological system containing an aquatic environment (water and sediment) and aquatic organisms (invertebrates, aquatic plants and fish) was built to simulate the natural rearing conditions. A single dose of $0.3 \, \mathrm{mg \, kg^{-1}}$ body weight of IVM was given to the fish by oral gavage. Water, sediment, the roots and leaves of the aquatic plants, the soft tissue of the invertebrates and the visceral mass and muscle of fish samples were collected at 0.5 hours, 1 day, 7 days, 15 days, 30 days, 45 days, 60 days and 70 days after the treatment. IVM concentration in each sample was determined using ELISA method.

**Results**. IVM was quickly and widely distributed into the whole aquatic system in one day, and then was highly accumulated in organisms resulting in long-term residues. IVM was exchanged multiple times between the different media, which caused continuous fluctuations in the concentration of IVM in the water and sediment. It was worth noting that there was a second peak value of IVM in the fish and invertebrates after 30 days. The environmental fate of the IVM in the aquatic micro-ecological system showed that the drug was transferred from the fish to aquatic plants in the first seven days, and then gathered in the water and sediment, finally accumulating in the invertebrates. Our results indicated that an effective aquatic micro-ecological system was successfully established, and it could be applied to the study the environmental fate of IVM, which will aid the scientific use of this anti-parasitic agent during aquaculture.

Corresponding author
Tongyan Lu, lutongyan@hrfri.ac.cn

## INTRODUCTION

With the increasing awareness of food and environmental security, public concern and scientific studies on pharmaceutical drugs in the environment have increased over the

previous years. Ivermectin (IVM) is a macrocyclic lactone derived from avermectins (AVMs), which is comprised of two homologues ($\geq$80% 22, 23-dihydroavermectin $B_{1a}$ and $\leq$20% 22, 23-dihydroavermectin $B_{1b}$) (*Rath et al., 2016*). As a class of broad-spectrum agents with the ability to kill endo- and ectoparasites (*Omura, 2008*), IVM has been primarily been used throughout the world to treat livestock (sheep, swine and horse) and pets to protect them against a broad variety of parasites only a few years after it was first made legal in 1981 (*Geary, 2005*). In particular, IVM (as Mectizan®) was employed to control and eliminate onchocerciasis for humans in poor rural communities in Africa and South America from 1987, and the Mectizan Donation Program administered 168 million treatments in 2013 (*Omura & Crump, 2014*).

Since its efficacy against sea lice infections in farmed Atlantic salmon (*Salmo salar* L.) without any treatment-associated host mortality (*Palmer et al., 1987*; *Johnson et al., 1993*), IVM has been widely used in the aquaculture industry (*Prasse, Lffler & Ternes, 2009*). Although the drug tolerance was species dependent (*Wu et al., 2012*), IVM had a narrow gap (between safe and toxic doses) in salmon and was highly toxic to freshwater aquatic species (*Garric et al., 2007*; *Ucan-Marin et al., 2012*). The degradation of IVM was very slow in aquatic environments, and the degradation rate in the sediment was only 28.3% after 70 days in a simulated river way environment (*Wu et al., 2012*), while its half-life in marine sediment was greater than 100 days (*Davies et al., 1998*). Due to its hydrophobic property and high affinity to organic matter (*Bloom & Matheson, 1993*), the long-term accumulation of IVM in the aquaculture environment was recognized as the diffusion sources of pollutants affecting ecosystems.

*Wall & Strong (1987)* reported that IVM could kill beneficial dung-degrading insects (*Coleoptera* sp. and *Scarabaeidae* sp.) when calves were treated with the recommended dose. In view of this situation, scientific researchers more closely examined the ecological fate and effects of IVM on the environment. In 2007, the standardized test methodology (mesocosm) of IVM potential environmental risk was created to evaluate the fate and exchange between water and sediment. The acute effects, chronic responses and long-term effects of IVM could be identified by this method (*Sanderson et al., 2007*). Following this, a test system, containing a cooling and water trap, was built to investigate the environmental fate of IVM in an aerobic sediment/water system. IVM could be rapidly sorbed to the sediment, converted into bound residues and transferred into several transformation products (TPs) (*Prasse, Lffler & Ternes, 2009*). In addition, the fate and effects of IVM on soil invertebrates in terrestrial model ecosystems were assessed in Terrestrial Model Ecosystems (TMEs), and the results showed that IVM generally had low to moderate effects on soil organisms (*Forster et al., 2011*). Moreover, *Rath et al. (2016)* found that IVM was difficult to desorb once sorbed to the soil, and the sorption parameters were dependent on the IVM concentration. IVM degradation by UV/$TiO_2$ and UV/$TiO_2$/$H2O_2$ was highly effective in water. All the studies described above were focused on the sorption, degradation and toxicity of IVM to the soil, sediment and invertebrates. Therefore, comparable studies in a complete aquatic system were merited and helped to elucidate the environmental fate and effects of IVM. In this study, we evaluated the fate of IVM in a simulated aquatic

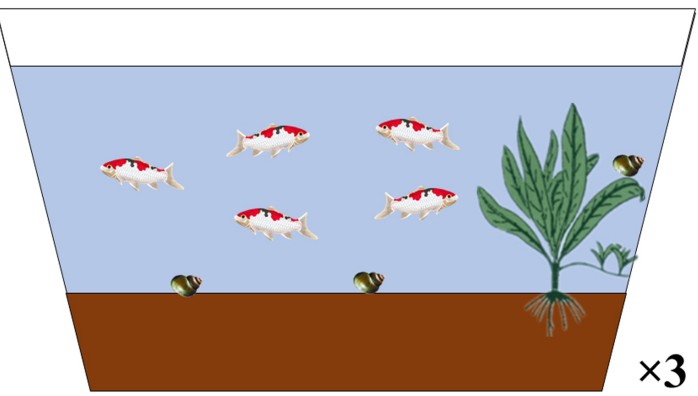

**Figure 1  Simulated aquatic micro-ecological system.** The system consists of water, sediment, brocarded carp, mudsnails and Amazon sword plants.

micro-ecological system containing an aquatic environment (water and sediment) and aquatic organisms (invertebrates, aquatic plants and fish).

## MATERIALS & METHODS

### Compound

IVM (99.5%) was purchased from Dr. Ehrenstorfer GmbH (Augsburg, Germany; Lot No. 10506). Acetonitrile and ethyl acetate (HPLC grade) were purchased from Merck KGaA (Darmstadt, Germany). Dimethyl sulfoxide (ACS grade) was purchased from Amresco (Cleveland, OH, USA). NaCl and $MgSO_4$ (analytical grade) were purchased from Aladdin (Shanghai, China). An avermectins ELISA Kit was purchased from Randox (Crumlin, United Kingdom; Cat No. AV3477).

### Construction of the simulated aquatic micro-ecological system

The study was conducted during the spring/summer at a temperature of 20–25 °C in Northeast China. The system was located in an open area without differentiated shading or wind exposure close to the aquaculture ponds. five cm thick sediment and 300 liters water, which were obtained from Hulan Aquaculture Experimental Station (Harbin, China), were added to a 400 liter polypropylene tank to build the simulated aquatic micro-ecological system. Three parallels were set up in the experiment. Forty brocarded carp (*Cyprinus carpio haematopterus*) with a mean body weight of 5.35 ± 0.48 g, 40 mudsnails (*Cipangopaludina cahayensis*) and 40 Amazon sword plants (*Echinodorus amazonicus*) with a mean length of 9–12 cm were placed into each system as shown in Fig. 1.

The system, with no IVM added or detected, was equilibrated for seven days in natural conditions. The water quality was tested daily and had a pH of approximately 7, while the oxygen level was >6 mg $L^{-1}$ due to the inflation pump. The water was replenished every two days.

## Treatment and experimental design

After the systems were stable, the experimental fish were subjected to IVM at a single dose of 0.3 mg kg$^{-1}$ body weight by oral gavage (*Yang, 2005*). The fish were starved at least for 24 h to ensure gut clearance before the oral administration. The samples, including the roots and leaves of the Amazon sword plant, the visceral mass and muscle of the brocarded carp, the soft tissue of the mudsnails, and the sediment and water ($n = 3$), were collected at the following time intervals respectively: 0.5 h, 1, 7, 15, 30, 45, 60 and 70 days. For water, two mL water from upper, middle and lower layer were sampled and mixed to be tested. For sediment, a one cm diameter casing was used to vertically take the bottom mud and mix it to be tested. For Amazon sword plants, three plants were took randomly and the soil around the roots was discarded. After that the leaves and roots were sampled respectively. For brocarded carp, three fish were collected randomly and the visecral mass and muscle were sampled. For mudsnails, three mudsnails were collected randomly and the shells were peeled off. The internal tissues were then sampled. All the samples were frozen immediately at −80 °C until assayed.

## Sample preparation

Water: A five mL water sample was centrifuged at 4,000 rpm for 10 min. three mL of supernatant was transferred to a new centrifuge tube. One volume ethyl acetate was added, and the mixture was vortexed for 10 min and centrifuged at 4 000 rpm for 5 min. The clear supernatant was transferred to a clean tube and dried using a stream of nitrogen gas. The residue was re-dissolved in 200 μL sample buffer (from the AVMs ELISA Kit) and vortexed for 2 min.

Sediment: four mL acetonitrile was added to 2 g sediment and homogenized for 10 min. A total of 0.05 g NaCl and 0.2 g $MgSO_4$ was then added. The mixture was immediately shaken to help reduce the development of aggregates and then centrifuged at 4,000 rpm for 12 min. The clear supernatant was transferred to a clean tube and dried using a stream of nitrogen gas. The residue was re-dissolved in one mL sample buffer and vortexed for 2 min.

Animal and plant: A 1 g sample of animal tissue (or a 2 g plant tissue sample) and four mL acetonitrile were mixed and homogenized. If the tissues were less than 1 g, samples were prepared by the ratio of 1:4 (sample quantity and acetonitrile). After 1 min extraction by vortexing, 0.05 g NaCl and 0.2 g $MgSO_4$ were added to the mixture and immediately shaken. The mixture was then centrifuged at 4 000 rpm for 12 min. All the liquid in the acetonitrile layer was transferred to a clean tube, and 100 μL dimethyl sulfoxide was added to the tube. The mixture was then dried using a stream of nitrogen gas. The residue was re-dissolved in 900 μL sample buffer (or 300 μL for the plant sample) and vortexed for 2 min.

## Ivermectin determination

An AVMs ELISA Kit was used to quantitatively measure the IVM in all the samples. The operating steps were performed using the manufacturer's instructions (Batch Number: 289106 and 289120). The concentration of IVM was calculated based on a standard curve using the corresponding coefficient.

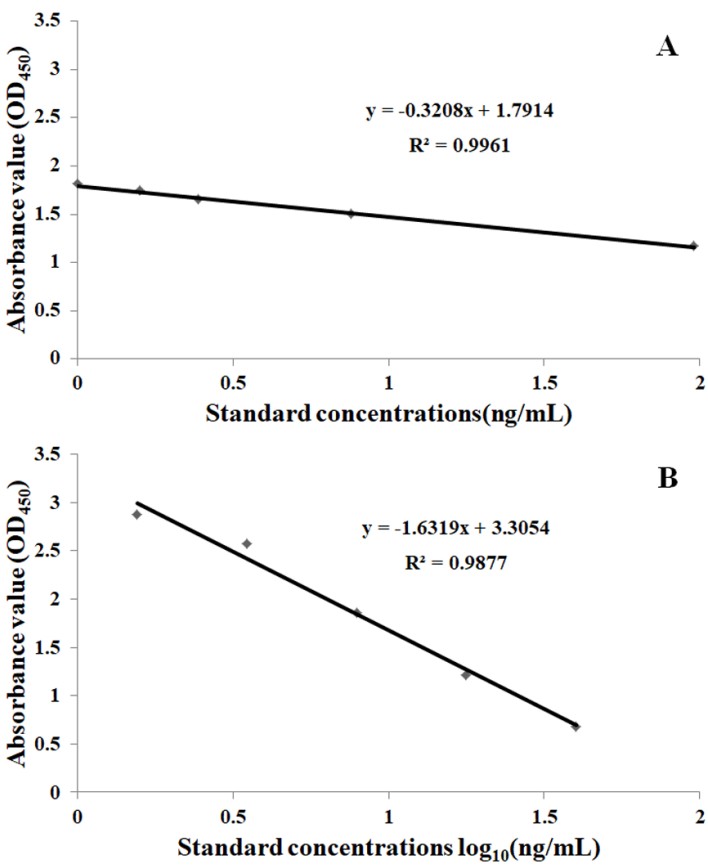

**Figure 2** **The standard curve of IVM at different standard concentrations.** (A) The low level standard curve. The standard curve with a $y$-axis value of $OD_{450}$ and an $x$-axis value of the standard concentration (0, 0.20, 0.39, 0.88, 1.98 ng/mL). (B) The high level standard curve. The standard curve with a $y$-axis value of $OD_{450}$ and an $x$-axis value of the standard concentration against the $log_{10}$ (1.56, 3.51, 7.90, 17.78, 40.00 ng/mL).

## Data processing

Data of the IVM concentration in the different samples were first standardized using min-max method (*Tan, Steinbach & Kumar, 2005*), and a single radar chart was created using the ggRadar function in the ggiraphExtra package of the R statistical software (*R Core Team, 2016*).

## RESULTS

### Establishment of the standard curves

According to the results of preliminary experiments, two types of AVMs ELISA kits (low and high levels) were chosen to detect the samples at the different concentration levels. Two representative standard curves for the quantification of IVM are shown in Fig. 2.

The regression equation of the low level standard curve (Batch Number: 289106) with a $y$-axis value of $OD_{450}$ and an $x$-axis value of the standard concentration (0, 0.20, 0.39, 0.88, 1.98 ng mL$^{-1}$) was $y = -0.3208x + 1.7914$ ($R^2 = 0.9961$). The regression equation of

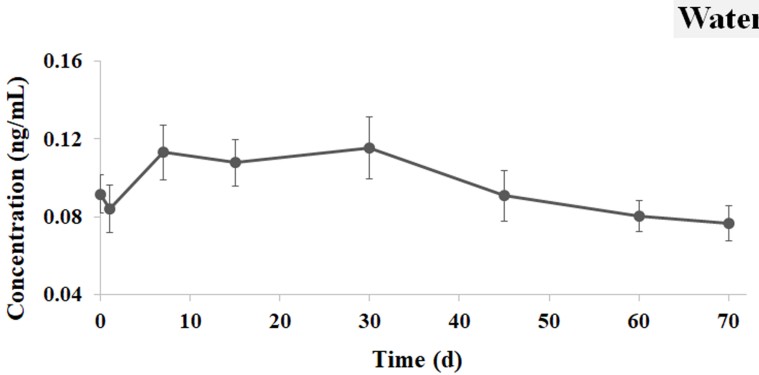

**Figure 3 IVM concentrations in the water.** The curve indicates the change of IVM concentrations in the water from 0.5 h to 70 d after the single oral administration.

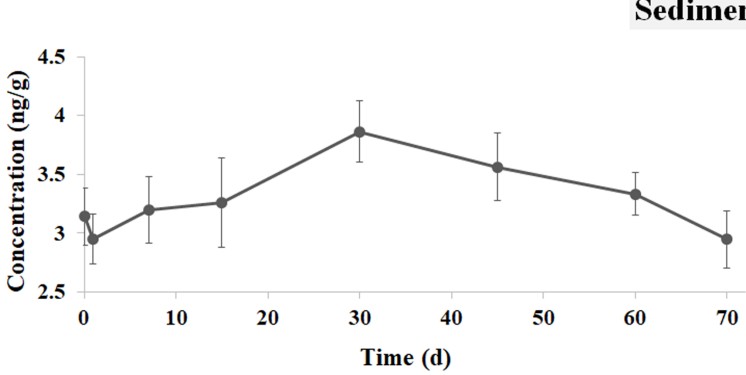

**Figure 4 IVM concentrations in the sediment.** The curve indicates the change of IVM concentrations in the sediment from 0.5 h to 70 d after the single oral administration.

the high level standard curve (Batch Number: 289120), with a $y$-axis value of $OD_{450}$ and an $x$-axis value of standard concentration against the $\log_{10}$ (1.56, 3.51, 7.90, 17.78, 40.00 ng mL$^{-1}$) was $y = -1.6319x + 3.3054$ ($R^2 = 0.9877$).

## Distribution of Ivermectin in the water and sediment

The concentration curve of IVM in the water is shown in Fig. 3. The concentration of IVM was 0.092 ng mL$^{-1}$ at 0.5 h after oral administration. It decreased to 0.084 ng mL$^{-1}$ one day later. The concentration reached its peak at 7 d (0.113 ng mL$^{-1}$) and 30 d (0.115 ng mL$^{-1}$). The concentration of IVM then gradually declined and reached a value of 0.076 ng mL$^{-1}$ at 70 d.

The concentration of IVM in the sediment reached 3.141 ng g$^{-1}$ at 0.5 h and accumulated continuously to its peak of 3.863 ng g$^{-1}$ at 30 d. After that, the IVM concentration in the sediment also gradually declined and reached a value of 2.946 ng mL$^{-1}$ at 70 d, which was similar to the trend with IVM in the water (Fig. 4).

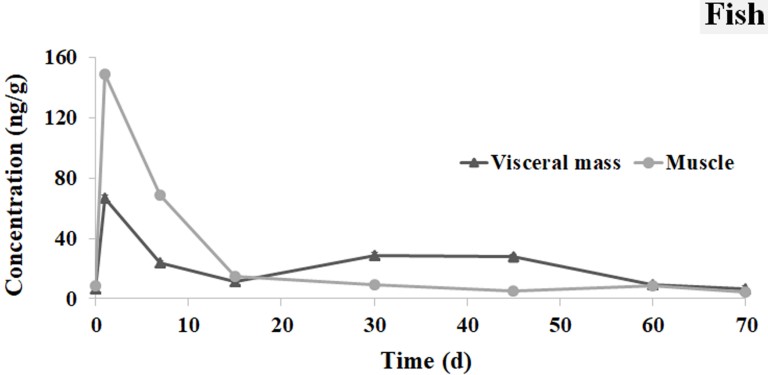

**Figure 5  IVM concentrations in the visceral mass and muscle of brocarded carp.** The curve indicates the change of IVM concentrations in the visceral mass and muscle of brocarded carp from 0.5 h to 70 d after the single oral administration.

## Distribution of Ivermectin in the brocarded carp

As shown in Fig. 5, the concentration of IVM in the muscle was 6.416 ng g$^{-1}$ at 0.5 h after the oral administration, and the peak appeared at 1 d with an IVM concentration of 67.080 ng g$^{-1}$. The concentration of IVM in the visceral mass was 218.613 ng g$^{-1}$ at 0.5 h. The concentration of IVM in the visceral mass declined sharply during the next two weeks. Its concentration reached 4.469 ng g$^{-1}$ in the muscle and 6.683 ng g$^{-1}$ in the visceral mass at 70 d. It is worth noting that there was a second peak in the visceral mass between 30 d and 45 d with a value of 27.796 to 28.979 ng g$^{-1}$.

## Distribution of Ivermectin in the Amazon sword plant

The concentration curves of the IVM in the leaves and roots of the Amazon sword plants are shown in Fig. 6. The concentration of IVM in the plant leaves was 19.573 ng g$^{-1}$ at 0.5 h, and it then decreased to 14.397 ng g$^{-1}$ at 7 d. At 15 d, the IVM concentration increased to 18.581 ng g$^{-1}$, and then remained relatively stable until 60 d. At 70 d, the IVM decreased to 14.040 ng g$^{-1}$ in the leaves.

Compared to the leaves, the concentration of IVM in the plant roots exhibited a different trend, which reached a peak value of 38.584 ng g$^{-1}$ at day 7. An obvious decline was observed from day 7 to 15, and IVM decreased to 28.622 ng g$^{-1}$. Finally, the IVM concentration decreased to 21.140 ng g$^{-1}$ at 70 d.

## Distribution of Ivermectin in the mudsnails

The concentration curve of IVM in the mudsnails is shown in Fig. 7. The concentration of IVM was determined to be 13.221 ng g$^{-1}$ at 0.5 h and then reached a peak value of 24.987 ng g$^{-1}$ at 7 d. A decline to 13.441 ng g$^{-1}$ was observed from 7 d to 15 d. From day 15 to 70, elimination of the IVM was very slow, and the concentration remained approximately 13 ng g$^{-1}$.

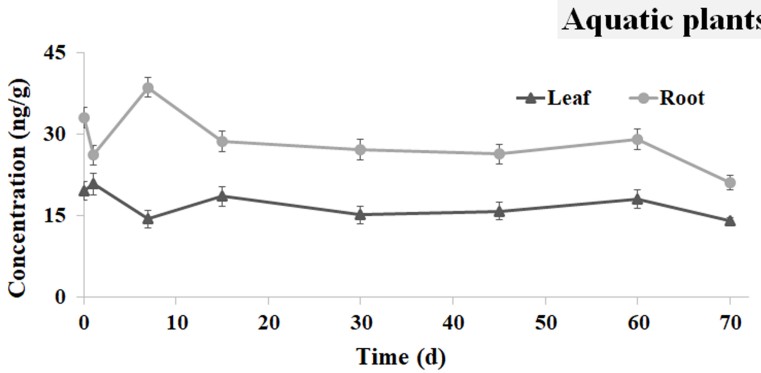

**Figure 6 IVM concentrations in the Amazon sword plant.** The curve indicates the change of IVM concentrations in root and leaves of the Amazon sword plants from 0.5 h to 70 d after the single oral administration.

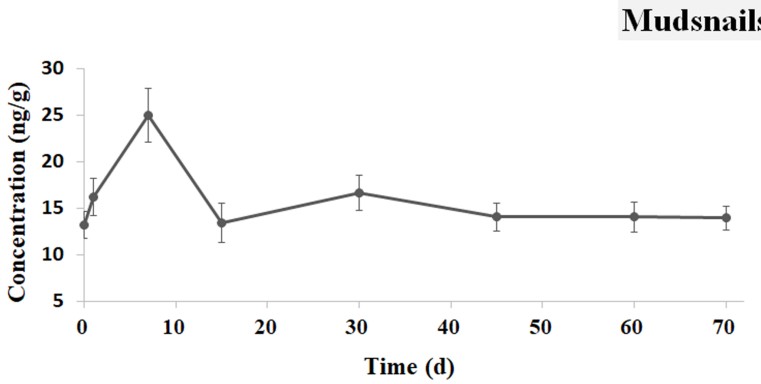

**Figure 7 IVM concentrations in the mudsnails.** The curve indicates the change of IVM concentrations in the soft tissue of the mudsnails from 0.5 h to 70 d after the single oral administration.

## Data analysis

As described above, the concentration of IVM in the samples was completely different after treatment. Thus, for comparison, the drug concentrations in seven types of samples at eight points in time were standardized and presented as a radar chart (Fig. 8).

With the extension of the administration time, IVM was transferred between different agents in the aquatic micro-ecological system. In one day, IVM appeared in the visceral mass of the brocarded carp and the leaves of the Amazon sword plant. There were many exchanges between the different agents from 1 to 7 days. The drug then transferred to the water and sediment over the following days until the 60th day. IVM finally appeared in the mudsnails.

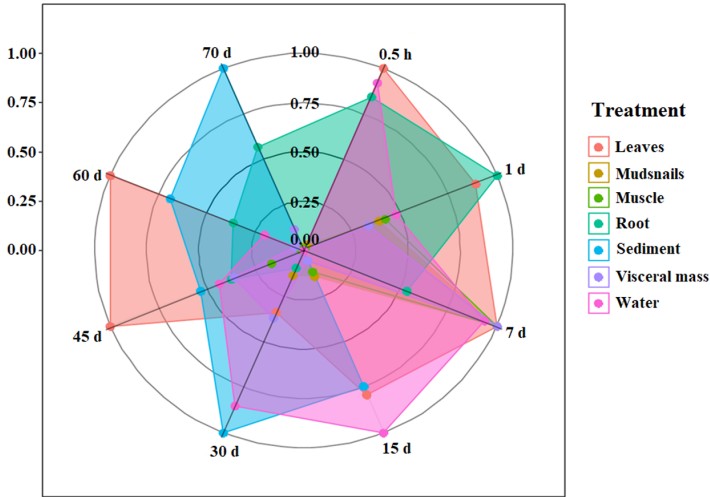

**Figure 8** Radar chart for the IVM concentrations in the aquatic micro-ecological system (consisting of water, sediment, fish, plants and mudsnails).

## DISCUSSION

### Methods and micro-ecological system for the detection

Oral administration was chosen to study the environmental fate of IVM in an aquatic system to ensure that the experimental fish were dosed with the accurate amount. As a class of potent anti-parasitic agent, the effective treatment dose of IVM is only 0.3 mg kg$^{-1}$ body weight by oral administration in freshwater aquatic organisms.

The immunochemical method was able to detect low levels of residue in the water, soil, and plant and animal samples (*Krotzky & Zeeh, 1995*). ELISA is a type of easily used, rapid, sensitive and specific method to quantitatively analysis of ultra micro amounts (*Dixon-Holland, 1992*; *Khalil et al., 2011*). So far, ELISA had been widely used to detect IVM (*Katharios, Pavlidis & Iliopoulou-Georgudaki, 2004*; *Shi et al., 2006*; *Menozzi et al., 2015*; *Bernigaud et al., 2016*).

Based on the culture pond, the constructed micro-ecological system simulated a natural environment to study the fate of IVM during aquaculture. The results indicated that this system works well and data obtained have a certain guiding significance to drug usage in the cultured ponds. However, there are still some shortages of the system. It is a small scale farming system and the experimental animals are small in size, which make it suitable for the research on drug fate in water environment. And the pharmacokinetic study cannot be carried out for the internal organs in this system.

### Distribution of IVM in the aquatic environment

In this study, the reason that the concentration of IVM in the sediment was higher than that in the water should be due to the hydrophobic properties of IVM and its high affinity to organic matter (*Bloom & Matheson, 1993*). After treatment, the IVM was quickly detected in both the water and sediment. This is consistent with reports that the IVM could rapidly diffuse from the water phase to the sediment particles (*Löffler et al., 2005*).

In addition, two concentration peaks of IVM appeared in the water and sediment. First, it appeared at 0.5 h after treatment. When the drug was given, part was immediately excreted by the gut to the aquatic environment for the nervous swimming of the fish and their gut cleaning. It was similar to the report that only 30% of IVM orally administered in salmon was measured in the muscle, blood, kidney and liver (*Høy, Horsberg & Nafstad, 1990*). After that, the second peak appeared at 30 d in both water and sediment, followed by a decline in the concentration of IVM in the aquatic organisms (including fish, mudsnails and plants). Thus, we deduced that both the water and sediment were crucial agents for the transfer of IVMs in aquatic system. In addition, IVM accumulated in the aquatic environment for a long time.

The IVM sorption to the sediment played a key role in its environmental fate (*Prasse, Lffler & Ternes, 2009*). In this study, IVM was quickly found in the sediment, and it remained for a long time (more than 70 days), which was similar to previous studies (*Davies et al., 1998*; *Mougin et al., 2003*; *Wu et al., 2012*). The reason for its persistence could be due to its rapid sorption and difficultly in desorbing from the soil (*Rath et al., 2016*). The long-term accumulation of IVM in an aquaculture environment was recognized as the diffusion sources of pollutants affecting ecosystems for its direct damage on non-target organisms and potential negative impact to sensitive ones (*Burridge et al., 2010*). The deposition of IVM will be considered to be cumulative over the period of excretion and deposition, and the levels found in the sediment will represent the cumulative total at the end of the deposition following treatment (*Davies et al., 1998*).

## Distribution of IVM in organisms

Biocondensation refers to the ability of organisms to attain high-level concentrations of chemicals through transportation and accumulation in the food chain. The concentrations of chemicals obviously increase when it accumulates in a type of organism. In this study, we detected the transfer of IVM between several different types of organisms, including fish, plants and mudsnails in the closed simulated aquatic micro-ecological system.

For oral administration, the concentration of IVM in the visceral mass was higher than in the muscle during the first days. The concentrations of IVM in the muscle accumulated quickly and reached a peak at 1 d. This result was similar to the pharmacokinetics research on IVM in *Salvelinus leucomaenis* (*Han et al., 2014*) and *Oncorhynchus mykiss* (*Kang et al., 2015*), which indicated that IVM had a secondary or multiple accumulation peaks in plasma, liver and kidney in 72 h after oral administration or i.p. injection. It was thought to be caused by multiple absorption of the intestinal-hepatic circulation and the gastrointestinal tract. However, this phenomenon is completely different from the mechanism caused by environmental drug exchange and accumulation after 30–45 days in this study. This phenomenon might be caused by the biocondensation of IVM from the water and sediment to the main organisms in the system. It was different from the one peak found in terrestrial animals (*Vokřál et al., 2019*). So far, there have been no reports on the occurrence of secondary accumulation peaks in organisms caused by long-term residues in aquaculture environments.

*Chen et al. (2007)* reported that *E. amazonicus* could adsorb substantial amounts of Ciprofloxacin (CPFX), especially by its leaves. IVM also appeared in the leaves of *E. amazonicus* within a short time in this study, which indicates that the leaves of *E. amazonicus* can adsorb some drugs. IVM accumulated to a higher level in the roots than in the leaves of the aquatic plants. It was hypothesized that the drug in the aquatic plants was primarily from the sediment because the concentration of IVM was higher in the roots than in the leaves.

Mudsnails played an important role in regulating the structure of an aquatic ecological system (*USEPA, 2009*), and they are usually used as important biological indicators of ecological toxicology (*Volker et al., 2014*; *Liu et al., 2015*) In this study, IVM appeared at 0.5 h and remained at a high level in the mudsnails for more than 70 days. That may be related to the drug residue in the sediment, which served as the food for the mudsnails (*Guo & Lin, 1997*). The concentration of IVM in the mudsnails, which can absorb pollutants from water bodies (*Cai et al., 2013*), could indicate the drug level in the whole aquatic system.

## CONCLUSIONS

In summary, we simulated an aquatic micro-ecological system and evaluated the fate of IVM in the environment and several types of organisms in this study. IVM could accumulate and be distributed in the water, sediment, fish, plants and mudsnails, and there was an obvious change of IVM concentration in different media over time.

### Funding

This work was supported by the Central Public-interest Scientific Institution Basal Research Fund, HRFRI (NO. HSY201705M), the Special Fund for Agro-scientific Research in the Public Interest (Grant No. 201203085) and the Central Public-interest Scientific Institution Basal Research Fund, CAFS (NO. 2017HY-ZD1009). The funders had no role in study design, data collection and analysis, decision to publish, or preparation of the manuscript.

### Grant Disclosures

The following grant information was disclosed by the authors:
Central Public-interest Scientific Institution Basal Research Fund, HRFRI: HSY201705M.
Agro-scientific Research in the Public Interest: 201203085.
Central Public-interest Scientific Institution Basal Research Fund, CAFS: 2017HY-ZD1009.

### Competing Interests

The authors declare there are no competing interests.

### Author Contributions

- Di Wang conceived and designed the experiments, performed the experiments, analyzed the data, contributed reagents/materials/analysis tools, prepared figures and/or tables, authored or reviewed drafts of the paper, approved the final draft.

- Bing Han conceived and designed the experiments, performed the experiments, analyzed the data, approved the final draft.
- Shaowu Li performed the experiments, analyzed the data, prepared figures and/or tables, authored or reviewed drafts of the paper, approved the final draft.
- Yongsheng Cao contributed reagents/materials/analysis tools, approved the final draft.
- Xue Du analyzed the data, approved the final draft.
- Tongyan Lu conceived and designed the experiments, approved the final draft.

## Data Availability

The raw measurements are available in the Supplemental File.

## Supplemental Information

Supplemental information for this article can be found online at http://dx.doi.org/10.7717/peerj.7805#supplemental-information.

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
