# Peer review of "Environmental fate of the anti-parasitic ivermectin in an aquatic micro-ecological system after a single oral administration"

_PeerJ, doi:10.7717/peerj.7805_

## Round 0.1 · original submission · Minor Revisions

Thank you for a well-prepared paper. The referees make some good suggestions to improve it. We look forward to a carefully revised paper.

Reviewer 1 ·

Basic reporting

No comment

Experimental design

No comment

Validity of the findings

No coomment

Annotated reviews are not available for download in order to protect the identity of reviewers who chose to remain anonymous.

Reviewer 2 ·

Basic reporting

no comment

Experimental design

no comment

Validity of the findings

no comment

Additional comments

As an effective anti-parasites drug, IVM has been widely used in the aquaculture industry. However, due to its hydrophobic property and high affinity to organic matter, the long-term accumulation of IVM in the aquaculture environment was recognized as the diffusion sources of pollutants affecting ecosystems. Finding a good bio-indicator to monitor the concentration of IVM in aquaculture environment will be benefit for subsequent operation to reduce the IVM residue level in environment. To this end, the authors in this manuscript established a system including an aquatic environment (water and sediment) and aquatic organisms (invertebrates, aquatic plants and fish) to mimic a real aquatic micro-ecological system. They measured the dynamic of IVM concentration in all different individuals included in this system after administrating IVM to fishes using gavage. They found that there was a second accumulation peak in fish and invertebrates post administration 30 days for first time. They also found that the mud snail can be used as an indicator for the concentration of IVM in the whole aquatic system as the in vivo IVM amount in mud snails remained at a high level more than 70 days. Because the system they established could be more similar with nature ecological aquatic system, it is a most important part of this manuscript that is different from previous studies which evaluated the environmental residue of IVM. The authors should emphasize this point in discussion part and also need to point out the shortage or disadvantage of this mimic system they established to measure IVM residues in environment.
In general, the methods and results in this study are reliable and meaningful to the management of drug residue in aquaculture. However, some language issues in the manuscript let the paper difficult to read and understand. The manuscript would benefit from a critic review by the authors on English grammar and style issues.
In addition, some minor points and remarks have been addressed to shed light on the manuscript.
1. IVM was found to accumulate in visceral mass twice, i.e. at 1 d and between 30 d and 45 d after administration, respectively. Why there was no second peak in muscle tissue? How to explain that?
2. Secondary accumulation of IVM was found in organisms. Is it found at first time? Is there any reference to show similar discovery? Or is there any drug else showed the similar accumulation characters in fishes and other organisms?
3. The “(Wall et al., 1987)” in line 64 should be deleted.
4. The “minutes” in lines 115-132 should be replaced by “ min”
5. The “ (Chen et al., 2007)” in line 246 also should be deleted
6. line 254, “hour” should be “h”

---

## Round 0.2 · Minor Revisions

Thank you for the revisions and response to referees comments. Please amend the figures as follows:
fig1 - nice
fig2 - standardise scales (same maxima and intervals) on graphs, remove frame lines around graphs, and put B below A to make best use of space (so can be larger), correct formatting erorrs in legend.
Figures 3 to 7 - the time axis must be evenly scaled (at present interval sizes vary) and this will affect the slopes of the lines.
Fig 8 - legend needs to explain the scale on the left, and numbers around the radar plot.

---

## Round 0.3 · Minor Revisions

The figures look the same. You cannot connect points by a line on a graph when the axis is not to a standard scale.

---

## Round 0.4 · accepted · Accept

Thank you for correcting the figures, and considering PeerJ for your good work.